# Gender Differences in the Prevalence and Correlates of Elder Abuse in a Community-Dwelling Older Population in Korea

**DOI:** 10.3390/ijerph16010100

**Published:** 2019-01-01

**Authors:** Gyeong-Suk Jeon, Sung-Il Cho, Kyungwon Choi, Kwang-Sim Jang

**Affiliations:** 1Department of Nursing, Division of Natural Science, Mokpo National University, Muan-gun 58554, Korea; gsj64@naver.com; 2Institute of Health and Environment, Seoul National University, Seoul 08826, Korea; 3Graduate school of Public Health and Institute of Health and Environment, Seoul National University, Seoul 08826, Korea; persontime@hotmail.com; 4Department of Nursing, Korea National University of Transportation, Chungbuk 27909, Korea; 5Department of Nursing, Dong-a College of Health, Yeongam-gun 58439, Korea; mktb45@hanmail.net

**Keywords:** elder mistreatment, gender differences, elder abuse

## Abstract

Background: We examined gender differences in the prevalence estimates and correlates of elder abuse in a community-dwelling older population in Korea. Methods: We analyzed responses from the Living Profiles of Older People Survey (LPOPS), which comprises a nationally representative sample of non-institutionalized Korean older adults living in the community. A total of 10,184 older persons (4179 men and 6005 women) were included in the analysis. Results: The overall rate of elder abuse was 9.9%, and emotional elder abuse was the most frequent type. Of the men and women subjects, 8.8 and 10.6%, respectively, had experienced elder abuse. We observed significant gender differences in the correlates of elder abuse. Educational attainment was significantly associated with elder abuse in men but not in women. Poor self-rated health was significantly associated with elder abuse in women but not in men. Household income and relationship with children were significantly associated with elder abuse in both men and women. Conclusion: Our results suggest that the factors that make elderly persons vulnerable to elder abuse may differ by gender. A better understanding of the risk factors for elder abuse across genders will facilitate the development of elder abuse prevention strategies, practices, and policies.

## 1. Introduction

Elder abuse—defined as intentional actions inflicted on an elder to cause harm or create a serious risk of harm, or failure to protect the elder from harm or meet the elder’s basic needs [1]—is a violation of the fundamental right to safety and freedom from violence and requires urgent social action [2]. It is also an important public health problem, as it increases the risks of morbidity and mortality and of negative physical and mental health outcomes [3]. As the older adult population increases, elder abuse will become an increasingly pressing social problem.

The prevalence of elder abuse varies widely due to the lack of a consensus definition of elder abuse and its subtypes and methodological cultural differences [4]. For example, based on a meta-analysis of 52 studies from 28 countries, the pooled prevalence of overall elder abuse was 15.7% (95% CI 12.8–19.3%) [4]. On the basis of 34 population-based studies, the global prevalence of elder abuse was 10.0% (95% CI, 5.2–18.9%) [5]. In the European Report on Preventing Elder Maltreatment [6], the prevalence of abuse in community-dwelling elders was approximately 3%. A systematic review [7] estimated the prevalence of elder abuse in Asia to be 0.022 to 62%. A recent meta-analysis suggested that higher prevalence estimates of elder abuse are associated with convenience sampling and a smaller sample size [4].

Studies in Korea have reported various prevalence estimates of elder abuse. The prevalence of elder abuse in Korean community settings is reported to range from 6.3% [8] to 6.6% [9]. Over the past few decades, research on elder abuse and mistreatment in Korea has increased substantially. However, most studies of elder abuse in Korea to date have focused on older people living in urban areas using a non-random sampling method; few have analyzed a nationally representative sample [10]. The results of these studies may not reflect the actual prevalence of elder abuse because of their small sample size and the biased characteristics of the participants, such as the degree of dependency. Kong and Jeon [11] used nationally representative data to examine the relationship between functional decline and elder abuse, but only assessed emotional abuse. An enhanced understanding of elder abuse is needed to identify risk and protective factors and devise prevention and intervention strategies. Therefore, an investigation of the prevalence and correlates of elder abuse using nationally representative data is needed.

Among the factors reportedly associated with elder abuse are age, female gender, lower socioeconomic status, marital status, and health status [12,13]. Although the majority of studies showed the importance of gender, most focused on definitions and subtypes; gender differences in elder abuse have been neglected [4]. According to meta-analyses of elder abuse [4,5], older women, especially those living in non-Western countries such as in Asia and the Eastern Mediterranean, are more likely to be abused, whereas the rate of elder abuse does not differ by gender in Western developed countries. There are gender differences in various health outcomes as well as in the degree of exposure to psychosocial, structural, and behavioral determinants of health [14]. Researchers have suggested that these gender difference in health and exposures to health determinants are associated with gender-related social roles [15]. Moreover, variation of gender-related social roles among countries may contribute to variations of gendered patterns on health. However, this has not been evaluated empirically. Therefore, we could hypothesize that the prevalence of, and risk factors for, elder abuse differ by gender. 

Dependency has been regarded as a risk factor or precursor for elder abuse, because elder abuse may occur as a result of caregiver stress which caring for dependent person may cause [16]. There has been considerable debate about gender difference in dependency [17]. However, in consideration of East Asian cultures traditionally based on patriarchy and Confucianism value, we suppose dependency may differ by gender in Korean elders. Therefore, research into all aspects of elder abuse in both genders is needed to develop gender-tailored elder abuse prevention and intervention strategies in countries with prominent social gender roles.

In this study, we evaluated the gender differences in the prevalence of overall and five categories of elder abuse and identified the predictors among the Korean population aged 65 years and older. We also explored gender differences in the correlates of elder abuse.

## 2. Methods

### 2.1. Design and Study Population

We used data from the cross-sectional Living Profiles of Older People Survey (LPOPS), which was conducted by the Korean Ministry of Health and Welfare in 2014. The LPOPS surveys have been conducted at three-year intervals since 1998 using nationally representative samples of non-institutionalized Korean older adults living in the community. The appropriate sample size was calculated using the 2010 population and housing census data, and a nationwide probability sample of older adults was selected using a stratified two-stage cluster sample design in 16 regions (7 metropolitans and 9 provincial regions) as well as in urban and rural areas. A total of 10,451 older adults completed interviews conducted by trained interviewers. Participants whose proxy responded or who did not respond to major questions on, for example, depressive symptoms, cognitive function, and income were also excluded from the analysis. Additionally, participants who had never married were excluded because the majority of perpetrators in elder abuse have been family members, such as the spouse or adult-children [7,8]. Consequently, 4179 men and 6005 women aged 65 years or older were analyzed.

### 2.2. Assessment and Measurements

In Johannesen and LoGiudice’s systemic review of risk factors of elder abuse [18], important risk factors for elder abuse were environment (etc. living arrangements, residence, social support), relationships (etc. family disharmony, poor understanding), and conditions requiring dependency (etc. cognitive impairment, functional dependency, psychological problems). To identify the predictors of elder abuse, three groups of risk factors (socioeconomic variables, social ties, and health status) were employed as independent variables.

Elder abuse. We defined elder abuse as any act of physical or verbal abuse, financial exploitation, caregiving or financial neglect committed by a family member, friend, neighbor, acquaintance, or paid professional against a person 65 years or older during the most recent 12-month period. The following single items were served as sub-category for elder abuse, and used to assess each of five types of it: (1) physical abuse; Has anyone tried to hurt or harm (e.g., pushing, hitting, etc.) you? (2) verbal abuse; Has anyone put you down or made you feel bad (e.g., avoiding talking, disregarding opinion, pretending not to hear, complaining)? (3) financial exploitation; Has anyone taken your money or assets (e.g., house or stocks) without your consent? (4) caregiving neglect; Does someone in your family not care (e.g., no nursing, no hygiene, no meals, etc.) for you when you are sick? (5) financial neglect; Does someone in your family not visit you or not offer you a living allowance? Those who answered “yes” in one of the five items were considered as experiencing that type of elder abuse.

Socioeconomic variables. Socioeconomic status was assessed by evaluating the participants’ educational attainment, economic activity (employment status), annual household income, marital status and living arrangements. Educational level was classified as college or above, high school, middle school, or elementary school or uneducated. Total household income was divided by the square root of the number of household members and categorized according to the tertile distribution of all responses combined (lowest 33.3%, middle 33.3%, and highest 33.3%). Age was categorized into three groups (65–74, 75–84, and 85 years and older), and area of residence (urban or rural), religion (yes or no), and current employment status (yes or no) were also assessed. Marital status was categorized into three groups (married, widowed, and separated/divorced), and living arrangements was classified as couple only, living alone, living with married children, and living with unmarried children

Measures of social ties. The following three variables were evaluated to assess social ties: relationship with children, relationship with friends and neighbors, and social participation. Relationships with children and friends and neighbors were assessed by the question “How would you rate your relationship with your children (friends and neighbors)?” Participants rated the quality of their relationships with friends and neighbors on a scale from 1 (very good) to 5 (very bad) in response to each question. We categorized the responses to this question as “very good/good”, “fair”, and “bad/very bad.” Social participation was assessed by asking whether the subject engaged in any of the following social activities: friendships, hobbies, leisure-time activities, or political societies. A “yes” response to any social activity was considered as involvement in a social participation group.

Measures of health status. The self-rated health, physical function, cognitive function, and psychological health of the participants were assessed. Self-rated health was measured using the question “How would you rate your health in general?” We dichotomized the five responses to this question into healthy (“very good”, “good”, or “fair”) and unhealthy (“bad” or “very bad”). The Korean version of the Activities of Daily Living (K-ADL) and the Instrumental Activities of Daily Living scale (K-IADL) was employed to assess physical dependency [19]. Respondents were asked whether they needed assistance when performing seven daily activities (dressing, washing face/shampooing/teeth-brushing, bathing, eating meals, getting up and moving out of the room, using the toilet, and controlling urination and defecation) and 10 different instrumental activities (personal hygiene and grooming, housekeeping, preparing meals, making and receiving phone calls, managing money, taking medications as prescribed, use of transportation, shopping, and doing laundry). K-ADL/IADL scores were categorized as having no limitations in daily activities (K-ADL/IADL = 0) and having limitations in daily activities (K-ADL/IADL ≥ 1). Cognitive function was assessed using the Korean version of the Mini-Mental State Examination (K-MMSE), the validity and reliability of which have been established [20]. According to the conventional classification criteria [20], cognitive function was categorized into the following three groups: severe cognitive impairment (K-MMSE ≤ 17), mild cognitive impairment (18 ≤ K-MMSE ≤ 23), and normal cognitive function (K-MMSE ≥ 24). Depressive symptoms were assessed using the Korean version of the Geriatric Depression Scale-Short Form (SGDS-K), which was originally developed [21] and translated into Korean [22]. The SGDS-K is composed of 15 items with response options of “yes = 1” or “no = 0”. In this study, a score ≥8 was taken as indicative of significant depressive symptoms. The SGDS-K has satisfactory validity and reliability (Cronbach’s alpha of 0.90) [22].

### 2.3. Statistical Analysis

The data are expressed as the frequencies, weighted proportions, and means (±standard deviation [SD]) of the socioeconomic, health, and social ties characteristics by gender. The distributions of related factors were compared using chi-squared tests (Table 1). We conducted a logistic regression analysis to assess correlates of elder abuse (Table 2). We present all results separately for men and women, because there was a significant interaction for elder abuse between gender and the lowest education group in logistic regression analysis (*p* < 0.05). To examine differences between gender groups, we used the Wald chi-squared test to compare the logit coefficients of gender-specific models [23]; no significant collinearity was detected between any of the covariates. All statistical tests were conducted using IBM SPSS software v. 23.0 for Windows (IBM Corp., Armonk, NY, USA).

### 2.4. Ethics Statement

This study was approved by the Ethics Review Board of Mokpo National University, with which the authors are affiliated (20180807-SB-007-01).

## 3. Results

The characteristics of the participants, including their socioeconomic status, social relationships, and health status, are shown in Table 1. The mean age of the women (74.37 years) was greater than that of the men (72.82 years). Almost one-quarter of the participants (men, 22.6%; women, 24.1%) lived in a rural community. The women were more likely to be widowed, live alone, be less educated, have lower equivalent household income, and participate in economic activity than the men. The majority of the men (91.0%) and women (89.9%) reported having a good or fair relationship with their children. Almost half of the men and women reported having good relationships with their friends and neighbors. Women were more likely to participate in social activities, such as hobby clubs, friendship groups, and political societies, than were men. Women were more likely to report poor health and depressive symptoms than men, whereas men were less likely to have lower cognitive function and limitations of ADL/IADL than women.

Of the men and women, 8.8 and 10.6%, respectively, had experienced elder abuse (such as physical and psychological abuse, financial exploitation, and caregiving and financial neglect) in the most recent 12-month period (Table 2). More than three-quarters of those who experienced overall elder abuse (8.1%) experienced a single type. Women (2.1%) were more likely to have experienced two or more types of elder abuse than men (1.4%). Among the five subtypes of elder abuse, the most prevalent was psychological abuse in both men (6.6%) and women (7.7%), whereas physical abuse was rare in both men (0.1%) and women (0.2%). The proportions of women subjected to psychological abuse, caregiving neglect, and financial neglect were significantly higher than those of men (*p* < 0.05). Regarding the perpetrators of elder abuse (Table 3), caregiving (93.6%) and financial neglect (99.6%) were reported to be perpetrated by children. Friends, neighbors, and others were the major perpetrators of psychological abuse and financial exploitation.

The prevalence of elder abuse according to socioeconomic, social tie, and health characteristics is shown in the left-hand panel of Table 4. In older men, the prevalence of elder abuse was higher in those living in urban areas. Those who lived alone were more likely to experience elder abuse than were those who lived with others. A lower educational level, lower equivalent household income and economic activity were associated with elder abuse. Those who reported having poor relationships with their children and friends/neighbors, and participated in social activities had a higher frequency of elder abuse than their counterparts who reported good or fair relationships. As expected, poor self-rated health status and depressive symptoms were significantly associated with elder abuse. Limitations in ADL/IADL was not associated with elder abuse. As for older women, the prevalence of elder abuse was also higher in those living in urban areas. A lower educational level and lower equivalent household income, having poor relationships with their children and friends/neighbors, poor self-rated health status and depressive symptoms were also associated with elder abuse in women. However, unlike men, economic activity, participation in social activities were not associated, and limitations in ADL/IADL was significantly related with elder abuse in women.

The right-hand side of Table 4 presents the results of the multivariate logistic regression analysis assessing correlates of elder abuse across gender. As for men, living in a rural area was related to a lower risk of elder abuse than living in an urban area (OR = 0.64, 95% CI = 0.47–0.87). Living alone was significantly associated (OR = 1.59, 95% CI = 1.15–2.20), but, living with married and unmarried children was not associated with elder abuse. A lower level of educational attainment (elementary school or uneducated) was related to a significantly higher risk of elder abuse (OR = 1.68, 95% CI = 1.24–2.39) Older men in middle- (OR = 1.40, 95% CI = 1.03–1.90) and low- (OR = 2.01, 95% CI = 1.43–2.83) income families were more likely to report elder abuse than were men in high-income families. Having a fair or poor relationship with children was related to a 2.22- and 5.19-fold higher risk of elder abuse. Men (OR = 1.43, 95% CI = 1.07–1.91) with depressive symptoms were more likely to be at risk of elder abuse than the corresponding reference groups. Poor self-rated health was not associated with elder abuse (OR = 0.90, 95% CI = 0.69–1.16; *p* < 0.1).

As for women, living in a rural area and living alone were related to elder abuse (respectively, OR = 0.7, 95% CI = 0.53–0.85, OR = 1.80, 95% CI = 1.41–2.30). Unlike men, living with married and unmarried children was associated with elder abuse (respectively OR = 1.47, 95% CI = 1.10–1.98, OR = 1.43, 95% CI = 1.05–1.94). A lower level of educational attainment was not related, while the relationship with children was strongly associated with elder abuse; this relationship was significantly stronger for women than for men (*p* < 0.1). Having a fair or poor relationship with children was related to 2.93- and 7.92-fold among women. Those with depressive symptoms (OR = 1.42, 95% CI = 1.16–1.75) were more likely to be at risk of elder abuse than the corresponding reference groups. Poor self-rated health (OR = 1.22, 95% CI = 1.00–1.49) and limitations in ADL/IADL (OR = 1.57, 95% CI = 1.22–2.02) were associated with elder abuse in women. 

Comparing the logit coefficients for men and women by Wald chi-square statistics, gender differences in education (*p* < 0.05), equivalent household income (*p* < 0.1), relationship with children (*p* < 0.1), and self-rated health status (*p* < 0.1) were significant. Although the gender difference was not significant, with a higher risk of elder abuse among women (OR = 1.57, 95% CI = 1.22–2.02) but not men (OR = 1.30, 95% CI = 0.83–2.04).

## 4. Discussion

This is the first large-scale population-based study of a nationally representative sample to examine the prevalence of elder abuse and its correlates by gender in a community-dwelling setting in Korea.

The overall prevalence of elder abuse in this study (9.9%) was similar to the global estimate (10.0%) from population-based studies [5], lower than the global prevalence (15.7%) in the meta-analysis [4], and higher than that reported (3%) in the European Report on preventing elder abuse [6]. Yon et al. [4]. reported that studies using random sampling and those performed in high-income countries had lower prevalence estimates, whereas studies with medium and small sample sizes reported significantly higher prevalence estimates. The prevalence of elder abuse in Korea in this study was lower than those in China (36.2%) [24], Hong Kong (27.5%) [25], and Japan (34.9%) [26], but not Singapore (8.3%, Chokkanathan, 2018) [27]. In addition, the prevalence of elder abuse in this study was markedly higher than that reported by Oh et al. [8]. (6.3%) with a sample of community-dwelling older adults in Seoul in 1999. The higher rate of elder abuse in this study compared to that reported by Oh et al. [8]. may be due in part to the weakening of the traditional Korean values of respecting older adults and filial piety (Hyo) caused by industrialization and modernization, as well as the increase in the older population over time.

Older women were more likely to experience elder abuse than older men in this study, consistent with previous reports [13,16,28]. Older women are more likely to outlive their partners. This longevity increases the possibility of exposure to risk factors for elder abuse, such as loss of independence and cognitive impairment. For older women, elder abuse may be a continuation of intimate partner violence (IPV) into old age, in most reported cases of which women are the victims and men the perpetrators [5]. In a systematic review and meta-analysis [4], there was no significant difference in the prevalence of elder abuse between older women and men in Europe and the Americas, whereas gender differences have been reported by similar studies performed in the Asia–Pacific and Eastern Mediterranean regions. Gender differences or symmetry in abuse victimization must be considered in the context of the social and cultural milieus surrounding gender roles.

There is an association between perpetration of IPV and traditional gender roles [29], and traditional gender roles are linked to structural gender inequalities [30]. That is, patriarchal societies promote maintenance of the traditional male dominance over women, which is associated with IPV [29]. Older women and residents of non-Western countries were more likely to be abused than those living in Western countries due to differences in family structure [5]. Korean patriarchal values and traditions, which are rooted in Confucianism, have been suggested to contribute to the occurrence of IPV [31]. Further studies should evaluate gender differences in elder abuse according to the level of gender inequality and cultural gender roles.

The majority of perpetrators of elder abuse against men and women in this study were friends or neighbors, followed by others. This is in contrast to prior reports that most perpetrators are family members, such as the spouse or adult children [7,8]. This difference may be due to the broad questions used for screening elder abuse in this study. The caregiving neglect and financial neglect items inquired about experiences of elder abuse by a family member or caregiver. Also, the financial exploitation items asked specifically about elder abuse by any other person and focused on negative social behaviors, not illegal or criminal behaviors. Such broad questions may have resulted in capturing a greater variety of types of elder abuse. Laumann et al. [13] also reported a high frequency of elder abuse by others and used broad questions to screen for elder abuse. Alternatively, Koreans may be reluctant to expose family shame and acknowledge an abusive situation [32]. Older Korean women are significantly less likely to acknowledge abusive experiences than African–Americans and Caucasians [32]. This Korean cultural norm may even result in older adults reporting a non-family member as the perpetrator of elder abuse instead of accusing a family member.

The additional important finding of this study is that the following risk factors for elder abuse differ by gender: socioeconomic status (education and household income), relationship with children, and self-rated health. In other words, socioeconomic status was significantly associated with a higher risk of elder abuse in older men, and physical health status and relationship with children in older women. However, we found a significant interaction effect only between gender and the lowest education level. This result suggests that gender difference in prevalence and correlates of elder abuse may reflect in part to differential exposure to various living conditions patterned by social gender roles [14], while potentially higher susceptibility to elder abuse in the lowest educational group among men may also play a role. Men as the breadwinners tend to engage in marketable activities, whereas women tend to play a social role as a homemaker by engaging in domestic and supportive behaviors such as childcare, cooking, and sewing [33]. The loss of social roles in the family with age may result in losses of power and function in older adults [34], increasing their vulnerability to elder abuse. Our results in part support the social-exchange theory of elder abuse, in which the power, in terms of social status and personal resources (e.g., money, power, the ability to work or provide care to others), of elderly persons declines with age. This results in older persons engaging in unequal social exchanges within the family and being dependent on others to meet their basic needs [16].

An alternative explanation for the gender difference in the effect of physical health status on the risk of elder abuse is the higher rate of morbidity in women. Older women were more likely to have a poor self-rated health status (women 50.5% vs. men 34.4%) and limitations in ADL and IADL (women 15.3% vs. men 7.0%) than were older men. This may increase the risk of elder abuse in women [35]. 

Strikingly, the relationship with adult children was the most important risk factor in both older men and women. In Western studies, family disharmony and a poor or conflicting relationships with family members have been reported as risk factors for elder abuse [18,36]. However, few studies have addressed the relationship with adult children, likely due to cultural differences in the living arrangements of older adults in Western and Asian countries. Unlike in Western countries, in which older adults living with their adult children is uncommon, in East Asian countries co-residence with adult children is a desirable or preferred living arrangement. Moreover, adult children have an obligation to provide care and financial support to their aged parents. This is in line with prior reports from Asia that adult sons- and daughters-in-law are the primary perpetrators of elder abuse [8,25], compared to the spouse/partner in Western countries [12,13].

In this study, older men and women living alone were at greater risk of elder abuse than those living with others. Prior studies have reported inconsistent results regarding the relationship between living situation and elder abuse. In studies conducted in the United States and Europe [18,36,37], a shared living environment was a major risk factor for elder abuse because this increases the opportunities for contact, and thus also conflict and tension. In contrast, Asian studies reported that living alone was associated with an increased risk of elder abuse [24,38]. Therefore, the impact of living alone on the risk of elder abuse may differ according to cultural background. Lee et al. [38] reported that many Korean older persons assume that they will live with, and receive various types of care from, their adult children on the basis of Confucian values and norms; thus, older persons who live alone consider themselves to be neglected.

Impaired cognitive function is associated with an elevated risk of elder abuse as it increases the risk of caregiver burnout [7,18,36]. However, in this study, older adults with severe cognitive impairment were less likely to experience elder abuse, possibly due to the difficulty in obtaining accurate information about elder abuse of older adults with severe cognitive impairment.

This study has limitations that should be noted. First, the characteristics of caregivers that contribute to the risk of elder abuse were not considered; such information is critical for understanding elder abuse. Second, self-reporting by older adults may be subject to recall bias, and older adults could be more reluctant to report particular types and perpetrators of elder abuse. Third, not all types of elder abuse were analyzed; moreover, each included type of elder abuse was assessed by only one short question. Finally, the cross-sectional nature of the study prohibits drawing inferences about the causality of the relationships between the risk factors and elder abuse. A further longitudinal prospective study is needed to assess these relationships.

These limitations notwithstanding, this study has important implications. First, our data were nationally representative and were weighted by census estimates, increasing the generalizability of the findings. Second, the results of this study indicate that considerable number of older adults especially who have low household income or bad relationship with adult-children are at risk for elder abuse in Korea. Particular attention should be given to detect elder abuse in community, and interventions for the prevention of elder abuse should focus on giving financial supports or many opportunities to make money for their own, building cohesive family relationship, and developing family supportive services and programs. Third, this study provides insight into gender differences in the risk factors and prevalence of elder abuse in Korean older persons and shows that cultural factors play an important role in elder abuse. More efforts should be placed on research to improve understanding of gender-based abuse in community dwelling Korean elders, and gender based interventions such as counseling or empowerment program for the female victims.

## 5. Conclusions

The prevalence of elder abuse was higher among women than among men aged 65 years and older in Korea, and the correlates of elder abuse differed significantly by gender. Socioeconomic status (educational level and income) was strongly associated with elder abuse among older men, and relationship with children and health status among older women. Given the complexity of elder abuse, further longitudinal studies are needed to determine the gender-specific relationships between elder abuse and relevant variables, as well as the underlying mechanisms, in diverse populations and countries. Such studies will also facilitate the development of elder abuse prevention strategies, practices, and policies.

## Figures and Tables

**Table 1 ijerph-16-00100-t001:** Sociodemographic and health characteristics of community-dwelling men and women aged 65 years and older in Korea.

	Men		Women		Gender Difference	Total	
	N	%	N	%	χ^2^ or t (*p*)	N	%
	4179		6005			10,184	
Age	72.82	(±6.08)	74.37	(±6.72)	<0.001	73.72	(6.50)
65–74	2514	(65.2)	3348	(55.3)	<0.001	5862	(59.4)
75–84	1457	(30.1)	2297	(35.5)		3754	(33.2)
85+	208	(4.7)	360	(9.3)		568	(7.3)
Residency area							
Urban	2836	(77.4)	3983	(75.9)	0.088	6819	(76.6)
Rural	1343	(22.6)	2022	(24.1)		3365	(23.4)
Marital status							
Married	3626	(86.6)	2732	(44.2)	<0.001	6358	(61.9)
Widowed	401	(9.2)	3066	(51.9)		3467	(34.1)
Divorced/separated	152	(4.2)	207	(3.9)		359	(4.0)
Living arrangement							
Couple only	2713	(61.7)	2110	(33.1)	<0.001	4823	(45.1)
Living alone	423	(9.9)	2031	(32.7)		2454	(23.2)
Living with married (or grand) children	399	(10.3)	1065	(20.1)		1464	(16.0)
Living with unmarried children	644	(18.1)	799	(14.2)		1443	(15.8)
Education							
College or above	1504	(40.7)	628	(13.0)	<0.001	2132	(24.5)
High school	725	(16.8)	581	(10.5)		1306	(13.1)
Middle school	1314	(28.7)	2087	(34.6)		3401	(32.1)
Elementary school or uneducated	636	(13.8)	2709	(42.0)		3345	(30.2)
Equivalent household income							
High (1st 33.3%)	1444	(37.8)	1677	(30.3)	<0.001	3121	(33.4)
Middle (2nd 33.3%)	1485	(35.2)	1916	(32.1)		3401	(33.4)
Low (3rd 33.3%)	1250	(27.1)	2412	(37.6)		3662	(33.2)
Economic activity							
Yes	1639	(37.8)	1639	(22.7)	<0.001	3278	(29.0)
No	2540	(62.2)	4366	(77.3)		6906	(71.0)
Relationship with children							
Good	3060	(71.4)	4154	(66.6)	<0.001	7214	(68.6)
Fair	780	(19.6)	1323	(23.3)		2103	(21.8)
Poor	289	(7.6)	395	(7.6)		684	(7.6)
No children	50	(1.4)	133	(2.4)		183	(2.0)
Relationship with friends/neighbors							
Good	2328	(54.0)	3362	(54.6)	0.587	5690	(54.4)
Fair	1408	(34.3)	2044	(33.9)		3452	(34.0)
Poor	443	(11.3)	599	(11.6)		1042	(11.6)
Social activity participation							
No	2194	(53.7)	2270	(38.1)	<0.001	4464	(44.6)
Yes	1985	(46.3)	3735	(61.9)		5720	(55.4)
Depressive symptoms	4.57	(±4.41)	5.93	(±4.61)	<0.001	5.36	(±4.58)
No	3441	(82.0)	4453	(73.5)	<0.001	7894	(77.1)
Yes	738	(18.0)	1551	(26.5)		2289	(22.9)
Self-rated health							
Healthy	2672	(65.7)	2926	(49.5)	<0.001	5598	(56.3)
Unhealthy	1507	(34.3)	3079	(50.5)		4586	(43.7)
Limitation of ADL/IADL							
No	3863	(93.0)	5105	(84.7)		8968	(88.1)
Yes	316	(7.0)	900	(15.3)		1216	(11.9)
Cognitive function (K-MMSE scores)	25.22	(±3.88)	22.62	(±4.98)		23.71	(±4.73)
Normal cognitive function (24–30)	2955	(73.7)	2739	(48.9)	<0.001	5694	(59.3)
Moderate cognitive impairment (18–23)	1006	(21.7)	2253	(34.9)		3259	(29.4)
Severe cognitive impairment (≤17)	218	(4.5)	1013	(16.1)		1231	(11.3)

*p*-values for gender differences.

**Table 2 ijerph-16-00100-t002:** Prevalence and types of elder abuse among community-dwelling men and women aged 65 years or over.

	Men		Women		Gender Difference	Total	
	N	%	N	%	χ^2^ (*p*)	N	%
N=	4179		6005			10,184	
Elder abuse	339	(8.8)	580	(10.6)	<0.001	919	(9.9)
Single type	283	(7.4)	469	(8.6)	<0.001	752	(8.1)
Multiple type (Two or more types)	56	(1.4)	111	(2.1)		167	(1.8)
Physical abuse	8	(0.1)	13	(0.2)	0.789	21	(0.1)
Psychological abuse	249	(6.6)	429	(7.7)	0.036	678	(7.2)
Financial exploitation	11	(0.3)	20	(0.4)	0.472	31	(0.3)
Caregiving neglect	44	(1.1)	93	(1.8)	<0.001	137	(1.5)
Financial neglect	92	(2.3)	160	(3.1)	0.015	252	(2.8)

*p*-values for gender differences.

**Table 3 ijerph-16-00100-t003:** Distribution of Perpetrators of elder abuse among community-dwelling men and women aged 65 years or over.

	N	Types of Perpetrators (%)							
		Spouse	Children	Grand Children	Brothers/Sisters	Other Relatives	Friends/Neighbor	Social Work Personnel	Others
Physical abuse									
Men	8	-	-	25.0	-	-	50.0	-	25.0
Women	13	20.0	-	20.0	10.0	-	30.0	-	20.0
Total	21	14.3	-	21.4	7.1	-	35.7	-	21.4
Psychological abuse **									
Men	249	15.4	6.4	0.7	1.8	1.1	40.4	1.8	32.5
Women	429	5.1	15.2	1.3	1.1	0.2	48.8	1.3	27.0
Total	678	9.0	11.8	1.1	1.4	0.5	45.6	1.5	29.1
Financial exploitation **									
Men	11	-	18.2	-	18.2	0.0	36.4	-	27.3
Women	20	-	9.1	-	0.0	9.1	81.8	-	0.0
Total	31	-	12.1	-	6.1	6.1	66.7	-	9.1
Caregiving neglect *									
Men	44	2.1	91.7	-	6.3	-	-	-	-
Women	93	4.6	94.5	0.9	0.0	-	-	-	-
Total	137	3.8	93.6	0.6	1.9	-	-	-	-
Financial neglect									
Men	92	-	100.0	0.0	-	-	-	-	-
Women	160	-	99.5	0.5	-	-	-	-	-
Total	252	-	99.6	0.4	-	-	-	-	-

* *p* < 0.05; ** *p* < 0.01 for gender differences.

**Table 4 ijerph-16-00100-t004:** Prevalence of elder abuse according to sociodemographic and health characteristics, and odds ratios (95% confidence intervals) for elder abuse among older men and women.

	Men %	Women %	Gender Difference χ^2^(*p*)	Men OR (95% CI)	Women OR (95% CI)	Gender Difference
*Prevalence of elder abuse*	8.8	10.6	0.002			
Age						
65–74	8.6	10.6	0.002	1	1	
75–84	9.5	11.0		0.87 (0.68–1.13)	0.90 (0.73–1.11)	
85+	7.0	9.9		0.61 (0.33–1.13)	0.70 (0.49–1.01)	
Residency area	**	**	0.415			
Urban	9.5	11.7		1	1	
Rural	6.1	7.3		0.64 (0.47–0.87)	0.67 (0.53–0.85)	
Living arrangement	**	**				
Couple only	7.2	6.5	<0.001	1	1	
Living alone	18.8	14.5		1.59 (1.15–2.20)	1.80 (1.41–2.30)	
Living with married children	9.9	10.5		1.33 (0.91–1.95)	1.47 (1.10–1.98)	
Living with unmarried children/others	8.1	11.5		1.12 (0.80–1.55)	1.43 (1.05–1.94)	
Education	**	*				
College or above	6.8	11.3	<0.001	1	1	^‡^
High school	9.2	8.9		1.22 (0.87–1.71)	0.73 (0.50–1.07)	
Middle school	9.1	9.6		1.21 (0.96–1.63)	0.77 (0.52–1.04)	
Elementary school or uneducated	13.4	11.7		1.68 (1.24–2.39)	0.78 (0.57–1.07)	
Equivalent household income	**	**	0.051			
High (1st 33.3%)	5.6	8.0		1	1	^†^
Middle (2nd 33.3%)	8.4	9.0		1.40 (1.03–1.90)	0.96 (0.75–1.24)	
Low (3rd 33.3%)	13.8	14.1		2.01 (1.43–2.83)	1.35 (1.03–1.77)	
Economic activity	**					
Yes	7.2	9.7	<0.001	1	1	
No	9.8	10.9		1.14 (0.88–1.48)	0.93 (0.75–1.17)	
Relationship with children	**	**				
Good	5.4	5.4	0.009	1	1	^†^
Fair	13.2	16.3		2.22 (1.67–2.94)	2.93 (2.36–3.63)	
Poor	29.9	38.4		5.19 (3.71–7.25)	7.92 (6.11–10.26)	
No children	6.9	11.8		1.14 (0.28–2.46)	1.56 (0.90–2.70)	
Relationship with friends/neighbors	**	**				
Good	6.7	8.4	0.872	1	1	
Fair	9.5	11.4		0.89 (0.68–1.18)	0.91 (0.74–1.12)	
Poor	16.2	19.9		1.14 (0.79–1.63)	1.15 (0.87–1.51)	
Social activity participation	**					
Yes	7.6	9.5	<0.001	1	1	
No	10.2	11.3		1.29 (0.99–1.67)	1.14 (0.92–1.41)	
Depressive symptoms	**	**	<0.001			
No	7.3	8.2				
Yes	10.7	13.2		1.43 (1.07–1.91)	1.42 (1.16–1.75)	
Cognitive function (K-MMSE scores)	*					
Normal cognitive function (24–30)	8.2	10.4	<0.001	1	1	
Moderate cognitive impairment (18–23)	10.8	11.2		1.03 (0.79–1.36)	0.86 (0.69–1.08)	
Severe cognitive impairment (≤17)	8.3	10.1		0.58 (0.32–1.07)	0.64 (0.47–0.88)	
Self-rated health	**	**				
Healthy	7.8	8.0	<0.001	1	1	^†^
Unhealthy	10.7	13.2		0.90 (0.69–1.16)	1.22 (1.00–1.49)	
Limitation of ADL/IADL		**				
No	8.8	9.9	<0.001	1	1	
Yes	11.4	14.7		1.30 (0.83–2.04)	1.57 (1.22–2.02)	
Nagelkerke’s R^2^				0.131	0.170	
Hosmer & Lemeshow; X^2^ (*p*-value)				14.271 (0.075)	(0.733)	

* *p* < 0.05; ** *p* < 0.01 for differences among the levels of each variable; ^†^
*p* < 0.1; ^‡^
*p* < 0.05 by Wald chi-square statistics to test the difference in the coefficients between the gender models.

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
