# Peer review of "Gender Differences in the Prevalence and Correlates of Elder Abuse in a Community-Dwelling Older Population in Korea"

_ijerph, 2019, doi:10.3390/ijerph16010100_

Round 1
Reviewer 1 Report
This paper examines elder mistreatment (EM) in Korea using a nationally representative sample of older Koreans and, more specifically, examines potential gender differences in EM experiences. The paper examines an important and timely issue that would be of interest not just to Korean researchers and policymakers, but to gerontologists around the globe. I do have some critiques that I feel need to be addressed prior to recommending for publication, listed below:
Is the term “elder mistreatment” the standard in the literature? In the US, it is usually referred to as “elder abuse” but I realize that this may be a cultural standard. I ask because I believe the paper will be noticed/read more if it uses the terminology other researchers are using.
Pg 2, line 62 – I would not use the word “perpetrators” as this is not a term typically used to describe a predicting variable (I would just use “predictors”).
Pg 2 – In the section on the Design and Study Population, the authors mention the data utilized listwise deletion on variables of interest (i.e., they deleted cases which were missing values on depressive symptoms, cognitive function, etc.). One thing of note is that respondents who were never married were also excluded from the analysis. The rationale for this decision is not clear – why were never married Koreans excluded? It is a small number of people who were cut, yes, but I don’t see a reason why they were cut at all. A justification for this decision needs to be given.
Pg 3 – In the Assessment and Measurements section, I think the dependent variable (Elder Mistreatment) needs to be more clearly explained. Are these all single “yes/no” items that are ultimately being combined into one EM scale? Or are these all being assessed separately as individual EM variables? It becomes clearer as the reader gets further into the paper, but it needs to be a bit more explicit here (particularly: since each item will ultimately serve as a sub-category for EM, those sub-categories, like physical abuse or financial exploitation, should be outlined here too).
In the Assessment and Measurements section under the “socioeconomic variables” section, it appears there are some variables missing (i.e., variables not listed that appear later in the paper/later in the tables). As an example, I don’t see anything on living arrangement or marital status here – they should be included.
Also in the Assessment and Measurements section, it is stated in the “health status” paragraph that the K-IADL was used to assess physical disability, and that if a respondent indicated they required assistance for any of the items then they were categorized as being physically disabled. I am not familiar with the K-IADL (although I am familiar with US measurements) and am wondering: is it typical to categorize someone as physically disabled even if they only need assistance with 1 of the items (rather than several)? If so, I think a citation is needed to justify this scoring.
Table 3: it is not immediately clear that these are percentages that are presented (this becomes clearer in the text, but it should be explicit in the table as well).
Given the fact that this paper is focusing on gender differences in EM outcomes, why were interaction effects not tested for? A moderation analysis (e.g., regression with interaction terms included) may provide further proof of how the independent variables may correlate with EM differently based on gender.
While there is a line at the end of the Conclusions about potential impacts, I think the authors could strengthen the paper by providing some potential applications in the Discussion – in other words, now that we know there are gender differences in EM in the Korean population, what do we do with that information? How do we utilize this information in combating EM in Korea? This does not need to be a detailed, robust outline of tackling the problem of EM – just a general direction of potential application.
Author Response
Reviewer 1’s Comments
Comment #1 Is the term “elder mistreatment” the standard in the literature? In the US, it is usually referred to as “elder abuse” but I realize that this may be a cultural standard. I ask because I believe the paper will be noticed/read more if it uses the terminology other researchers are using.
►Response: Thank you for valuable comments. As you suggested, we replaced the term “elder mistreatment” with the term “elder abuse”.
Comment #2 Line 62 – I would not use the word “perpetrators” as this is not a term typically used to describe a predicting variable (I would just use “predictors”).
►Response: Thank you for pointing out our mistake. We have corrected it as you suggested
Comment #3 In the section on the Design and Study Population, the authors mention the data utilized listwise deletion on variables of interest (i.e., they deleted cases which were missing values on depressive symptoms, cognitive function, etc.). One thing of note is that respondents who were never married were also excluded from the analysis. The rationale for this decision is not clear – why were never married Koreans excluded? It is a small number of people who were cut, yes, but I don’t see a reason why they were cut at all. A justification for this decision needs to be given.
►Response: Thank you for pointing out the details. We have added the reason as you suggested (Line 77-79)
Comment #4 In the Assessment and Measurements section, I think the dependent variable (Elder Mistreatment) needs to be more clearly explained. Are these all single “yes/no” items that are ultimately being combined into one EM scale? Or are these all being assessed separately as individual EM variables? It becomes clearer as the reader gets further into the paper, but it needs to be a bit more explicit here (particularly: since each item will ultimately serve as a sub-category for EM, those sub-categories, like physical abuse or financial exploitation, should be outlined here too).
►Response: Thank you for pointing out this unclear sentence. We have corrected it more clearly (Line 91-99).
Comment #5 In the Assessment and Measurements section under the “socioeconomic variables” section, it appears there are some variables missing (i.e., variables not listed that appear later in the paper/later in the tables). As an example, I don’t see anything on living arrangement or marital status here – they should be included.
►Response: Thank you for pointing out the details. We have added them as you suggested (Line 108-110).
Comment #6 Also in the Assessment and Measurements section, it is stated in the “health status” paragraph that the K-IADL was used to assess physical disability, and that if a respondent indicated they required assistance for any of the items then they were categorized as being physically disabled. I am not familiar with the K-IADL (although I am familiar with US measurements) and am wondering: is it typical to categorize someone as physically disabled even if they only need assistance with 1 of the items (rather than several)? If so, I think a citation is needed to justify this scoring.
►Response: We totally agree with your comment. We used K-ADL/IADL to assess physical dependency (which means limitations in daily activities). We have corrected the sentences and added more details (Line 125-133).
Comment #7 Table 3: it is not immediately clear that these are percentages that are presented (this becomes clearer in the text, but it should be explicit in the table as well).
►Response: Thank you for pointing out the detail. We have added it in the table 3.
Comment #8 Given the fact that this paper is focusing on gender differences in EM outcomes, why were interaction effects not tested for? A moderation analysis (e.g., regression with interaction terms included) may provide further proof of how the independent variables may correlate with EM differently based on gender.
►Response: Thank you for valuable comments. We tested for interaction terms in logistic regression and found significant interactions for elder abuse between gender and education only. Therefore, we added about this results in statistical analysis of method section (line 148-150) and discussion section (line 294-298) of the revised paper. Previous test results for gender differences in odds ratios (significant level were 0.1 or 0.05) using Wald chi-square test proposed by Allison (1999) were kept for comparison.
The results of logistic regression analysis with interaction terms included are as follows;
Model 1 | Model 2 | Model 3 | Model 4 | ||||||||||||
OR | 95% CI | OR | 95% CI | OR | 95% CI | OR | 95% CI | ||||||||
Gender(Ref=Men) | |||||||||||||||
Women | 1.21 | 0.89 | 1.65 | 1.32 | 0.93 | 1.86 | 0.93 | 0.74 | 1.17 | 0.97 | 0.77 | 1.21 | |||
Age (Ref=65-74) | |||||||||||||||
75–84 | 0.89 | 0.76 | 1.04 | 0.90 | 0.76 | 1.05 | 0.89 | 0.76 | 1.04 | 0.89 | 0.76 | 1.04 | |||
85+ | 0.80 | 0.57 | 1.12 | 0.80 | 0.57 | 1.12 | 0.80 | 0.57 | 1.12 | 0.80 | 0.57 | 1.12 | |||
Residency area(ref=Urban) | |||||||||||||||
Rural | 0.89 | 0.75 | 1.05 | 0.89 | 0.76 | 1.05 | 0.89 | 0.75 | 1.05 | 0.89 | 0.75 | 1.05 | |||
Living arrangement (ref= Couple only) | |||||||||||||||
Living alone | 1.66 | 1.37 | 2.01 | 1.65 | 1.37 | 2.00 | 1.64 | 1.36 | 1.98 | 1.63 | 1.35 | 1.97 | |||
Living with married children | 1.29 | 1.02 | 1.64 | 1.31 | 1.03 | 1.67 | 1.30 | 1.02 | 1.65 | 1.30 | 1.02 | 1.65 | |||
Living with unmarried children/others | 1.22 | 0.96 | 1.54 | 1.22 | 0.96 | 1.53 | 1.21 | 0.96 | 1.52 | 1.21 | 0.96 | 1.52 | |||
Education (ref= College or above) | |||||||||||||||
High school | 0.92 | 0.70 | 1.20 | 1.02 | 0.72 | 1.45 | 0.93 | 0.71 | 1.22 | 0.93 | 0.71 | 1.22 | |||
Middle school | 0.92 | 0.74 | 1.15 | 1.01 | 0.75 | 1.36 | 0.94 | 0.75 | 1.18 | 0.94 | 0.75 | 1.17 | |||
Elementary school or uneducated | 0.97 | 0.76 | 1.24 | 1.24 | 0.88 | 1.76 | 0.98 | 0.76 | 1.25 | 0.98 | 0.76 | 1.25 | |||
Equivalent household income(ref=High) | |||||||||||||||
Middle (2nd 33.3%) | 1.10 | 0.80 | 1.50 | 1.01 | 0.82 | 1.24 | 1.01 | 0.82 | 1.24 | 1.01 | 0.83 | 1.25 | |||
Low (3rd 33.3%) | 1.58 | 1.15 | 2.17 | 1.33 | 1.07 | 1.66 | 1.33 | 1.07 | 1.65 | 1.33 | 1.07 | 1.66 | |||
Economic activity(ref=Yes) | |||||||||||||||
No | 0.97 | 0.81 | 1.15 | 0.97 | 0.82 | 1.15 | 0.98 | 0.83 | 1.16 | 0.98 | 0.83 | 1.16 | |||
Relationship with children (ref=Good) | |||||||||||||||
Fair | 2.42 | 2.02 | 2.89 | 2.42 | 2.02 | 2.89 | 2.15 | 1.62 | 2.85 | 2.42 | 2.02 | 2.88 | |||
Poor | 6.94 | 5.62 | 8.57 | 6.95 | 5.64 | 8.58 | 6.21 | 4.52 | 8.53 | 6.96 | 5.64 | 8.59 | |||
No children | 1.61 | 0.99 | 2.60 | 1.58 | 0.97 | 2.56 | 1.23 | 0.43 | 3.51 | 1.59 | 0.98 | 2.58 | |||
Relationship with friends/neighbors (ref=Good) | |||||||||||||||
Fair | 0.95 | 0.80 | 1.12 | 0.95 | 0.80 | 1.12 | 0.95 | 0.80 | 1.13 | 0.95 | 0.80 | 1.13 | |||
Poor | 1.29 | 1.03 | 1.62 | 1.29 | 1.03 | 1.62 | 1.31 | 1.04 | 1.64 | 1.30 | 1.04 | 1.64 | |||
Social activity participation(ref=Yes) | |||||||||||||||
No | 1.22 | 1.03 | 1.45 | 1.22 | 1.03 | 1.45 | 1.21 | 1.02 | 1.44 | 1.21 | 1.02 | 1.44 | |||
Depressive symptoms(ref=No) | |||||||||||||||
Yes | 1.47 | 1.24 | 1.75 | 1.47 | 1.24 | 1.75 | 1.47 | 1.23 | 1.75 | 1.47 | 1.24 | 1.75 | |||
Self-rated health(ref=Healthy) | |||||||||||||||
Unhealthy | 1.22 | 1.04 | 1.43 | 1.22 | 1.04 | 1.43 | 1.22 | 1.04 | 1.43 | 1.13 | 0.88 | 1.45 | |||
Limitation of ADL/IADL(ref=No) | |||||||||||||||
Yes | 1.39 | 1.12 | 1.74 | 1.40 | 1.12 | 1.74 | 1.39 | 1.12 | 1.74 | 1.39 | 1.11 | 1.73 | |||
Cognitive function (K-MMSE scores)(ref=Normal) | |||||||||||||||
Moderate dementia (18–23) | 0.91 | 0.76 | 1.08 | 0.92 | 0.77 | 1.09 | 0.91 | 0.76 | 1.08 | 0.91 | 0.76 | 1.08 | |||
Severe dementia (≤17) | 0.70 | 0.54 | 0.91 | 0.71 | 0.54 | 0.92 | 0.69 | 0.53 | 0.90 | 0.69 | 0.53 | 0.90 | |||
Gender X household income | |||||||||||||||
High (1st 33.3%) by gender | |||||||||||||||
Middle (2nd 33.3%) by gender | 0.86 | 0.58 | 1.29 | ||||||||||||
Low (3rd 33.3%) by gender | 0.75 | 0.52 | 1.10 | ||||||||||||
Gender X education | |||||||||||||||
College or above by gender | |||||||||||||||
High school by gender | 0.77 | 0.45 | 1.32 | ||||||||||||
Middle school by gender | 0.79 | 0.51 | 1.21 | ||||||||||||
Elementary school or uneducated by gender | 0.64 | 0.40 | 0.99 | ||||||||||||
Gender X relationship with children | |||||||||||||||
Good by gender | |||||||||||||||
Fair by gender | 1.20 | 0.85 | 1.70 | ||||||||||||
Poor by gender | 1.21 | 0.82 | 1.79 | ||||||||||||
No children by gender | 1.43 | 0.44 | 4.61 | ||||||||||||
Gender X self-rated health | |||||||||||||||
Healthy by gender | |||||||||||||||
Unhealthy by gender | 1.13 | 0.83 | 1.52 | ||||||||||||
Comment #9 While there is a line at the end of the Conclusions about potential impacts, I think the authors could strengthen the paper by providing some potential applications in the Discussion – in other words, now that we know there are gender differences in EM in the Korean population, what do we do with that information? How do we utilize this information in combating EM in Korea? This does not need to be a detailed, robust outline of tackling the problem of EM – just a general direction of potential application
►Response: Thank you for valuable comments. We have added more implications as you suggested (Line 346-355).
Reviewer 2 Report
Excellent initial work on this topic in this population. It is of particular important as Asian societies are seemed to be tolerant and respectful of the elderly compared to western countries. therefore this article is of extreme relevance to show an increased pattern of abuse towards aging population. My only recommendation would be to make sure the studies are cited correctly within the text. For example, the largest study for the United States and still currently use for prevalence of EM in the US (Acierno et al. 2010) is not cited correctly within the text but it is in the reference section. Make sure to add this reference particularly when your results are very similar in prevalence % to the NEMS.
Author Response
Reviewer 2’s Comments
Comment #1 Excellent initial work on this topic in this population. It is of particular important as Asian societies are seemed to be tolerant and respectful of the elderly compared to western countries. therefore this article is of extreme relevance to show an increased pattern of abuse towards aging population. My only recommendation would be to make sure the studies are cited correctly within the text. For example, the largest study for the United States and still currently use for prevalence of EM in the US (Acierno et al. 2010) is not cited correctly within the text but it is in the reference section. Make sure to add this reference particularly when your results are very similar in prevalence % to the NEMS.
►Response: Thank you for pointing out our mistake. We have added it as you suggested (Line 258)
Reviewer 3 Report
The research focuses on an important topic which has not received much attention in Asia. Below are my comments
Introduction
Introduction should critically bring out the need for studying abuse in the context of gender. There is a need to critically analyse the factors (i.e. socio-economic and structural inequalities that are disproportionately experienced by older women) that place older women at an increased for abuse. Dependency, though important, is not a consistent factor that is associated with abuse. Of course there are some information but it needs to be elaborated on.
Please do remove the statement ‘older women have stronger normative dependency than older men because they have learned to take for granted that they should remain attached to their parents before marriage, their husband after marriage, and any adult sons when they become older..’ It is value laden and blames the older women…
Methods
How was the elder mistreatment experience computed? What were the response format? ( yes/no or frequency of acts or ???).
There is a measure on cognitive impairments. For those with severe dementia/moderate dementia who were the respondents ( was it by proxy?)
Results
Rate of multiple mistreatments should be provided.
Separate logistic regression analysis has been considered for male and female older adults. Given that research is not clear whether odds ratio can be compared across different samples there is a need for a reference. Alternatively, the authors might consider writing results separately for male and female older adults, so that the readers are not misled. Following it the authors can describe the differences.
Discussion
It is well written. However some of the information should be placed in the introduction so that the readers are clear as why specific variables were included.
There is a need for a succinct implication .
It is a good research topic and is well written. The above are some pointers that the authors might consider to make it more relevant.
Author Response
Reviewer 3’s Comments
Comment #1 Introduction should critically bring out the need for studying abuse in the context of gender. There is a need to critically analyse the factors (i.e. socio-economic and structural inequalities that are disproportionately experienced by older women) that place older women at an increased for abuse. Dependency, though important, is not a consistent factor that is associated with abuse. Of course there are some information but it needs to be elaborated on.
►Response: Thank you for valuable comments. We have added the explanation of dependency as a risk factor in elder abuse as you suggested (Line 55-57).
Comment #2 Please do remove the statement ‘older women have stronger normative dependency than older men because they have learned to take for granted that they should remain attached to their parents before marriage, their husband after marriage, and any adult sons when they become older..’ It is value laden and blames the older women…
►Response: Thank you for valuable comments. We have rephrased the sentence not to be value laden as you suggested (Line 58-59)
Comment #3 How was the elder mistreatment experience computed? What were the response format? (yes/no or frequency of acts or ???).
►Response: Thank you for pointing out this unclear sentence. We have corrected it more clearly (Line 91-99).
Comment #4 There is a measure on cognitive impairments. For those with severe dementia/moderate dementia who were the respondents (was it by proxy?)
►Response: Thank you for pointing out the details. K-MMSE is a dementia screening test that can be easily applied in clinical and community settings. However, we included only the elders who answered for themselves. Therefore, the term “cognitive impairment” is more appropriate than “dementia” in this study. We have added explanation to the participants who were exclude in this study (Line 75-76), and replaced “dementia” with “cognitive impairment”.
Comment #5 Rate of multiple mistreatments should be provided.
►Response: Thank you for pointing out unclear expression. We have added the expression “multiple type” beside “two or more types” in the table 2.
Comment #6 Separate logistic regression analysis has been considered for male and female older adults. Given that research is not clear whether odds ratio can be compared across different samples there is a need for a reference. Alternatively, the authors might consider writing results separately for male and female older adults, so that the readers are not misled. Following it the authors can describe the differences
►Response: Thank you for valuable comments. We have revised them as you suggested (Line 193-233).
Comment #7 Some of the information should be placed in the introduction so that the readers are clear as why specific variables were included.
►Response: Thank you for valuable comments. We have added them as you suggested (Line 82-87)
Comment #8 There is a need for a succinct implication.
►Response: Thank you for valuable comments. We have added more implications as you suggested (Line 346-355).